# One-Dimensional Compressibility and Creep Characteristics of Unsaturated Compacted Loess Based on Incremental Loading and Constant Rate of Strain Methods

Pengju Qin [1,2,3,*], Qingchen Yan [1], Yu Lu [4], Chungang Yang [1], Zhiwei Song [1] and Chunbao Li [5]

1    College of Civil Engineering, Taiyuan University of Technology, Taiyuan 030024, China
2    Shanxi Transportation Technology Research and Development Co., Ltd., Taiyuan 030032, China
3    College of Mining Engineering, Taiyuan University of Technology, Taiyuan 030024, China
4    Department of Structural Engineering, University of California San Diego, La Jolla, CA 92093, USA
5    Department of Civil Engineering, China University of Petroleum (East China), Qingdao 266580, China
*    Correspondence: qinpengju@tyut.edu.cn

**Abstract:** In engineering practice, unsaturated compacted loess is often utilized as a filling material in the loess regions. The loess inevitably undergoes one-dimensional compressibility and creep deformation due to the long-term effects of the upper soil layers and buildings. When the deformation is large enough, it tends to damage buildings and threaten engineering safety. In this regard, the one-dimensional compressibility and creep properties of unsaturated compacted loess based on incremental loading (IL) and constant rate of strain (CRS) methods have been studied. First, soil materials with an initial moisture content of 15% were prepared and then compacted into soil samples with an 80 mm diameter and a 10 mm height. Second, the compressibility and creep properties of the compacted loess samples obtained via the IL and CRS compression tests were compared and analyzed. In this study, several parameters, including the primary compression index $C_c$ and secondary compression index $C_\alpha$, were derived. Meanwhile, the moisture content of the samples was measured via electrical resistivity methods. Finally, the microstructural characteristics were derived via nuclear magnetic resonance (NMR) and scanning electron microscopy (SEM) tests. The results showed that $C_c$ and $C_\alpha$ increased with the increase in moisture content and vertical stress; $C_\alpha/C_c$ ranged from 0.026 to 0.042. Compared with the compression parameters and deformation of the samples, those obtained via the CRS tests are a little larger than those obtained via the IL tests for a given loading and initial moisture content. The electrical resistivity depends on pore water-connected channels, which were deeply affected by the initial moisture content, vertical stress and loading duration (or strain rate). Moreover, as vertical stress increased, the pore size and pore area gradually decreased, the coarse particles were broken, and the fine particles increased. The contacts between particles changed from point-to-point contacts and edge-to-edge contacts to face-to-face contacts. Meanwhile, as vertical stress and loading rate increased, the loess particles were apt to vary from irregular elongated particles to equiaxial circular particles. This investigation can provide a theoretical base and experimental support for improving ground stability and preventing landslide disasters in loess regions.

**Keywords:** compacted loess; IL; CRS; compression index; resistivity; NMR; SEM

## 1. Introduction

Loess, a loose aeolian deposit of yellowish silt-sized dust, mainly formed during the Quaternary period and is widely distributed around the world. In China, loess is primarily found in the north and northwest areas. Due to the lack of rainfall and low water table in the northern regions, loess in these regions is often in an unsaturated state [1]. With the implementation of "Western Development" and "The Belt and Road" national strategy in China, the number of engineering constructions in the loess regions has gradually increased.

To increase the stability of grounds and roadbeds, the intact loess is often compacted via ramming or rolling in engineering practices [2]. Nonetheless, the roadbed is vulnerable to deformation and destruction due to long-term highway operation and increasing traffic [3]. In addition, landslide failure accelerates as the moisture content of the soil increases during rainfall, posing a significant threat to roads, rivers and human life [4,5]. In order to prevent the settlement of building and road structures and landslide creep failure in loess regions, it is critical to investigate the creep properties of unsaturated compacted loess.

There has been some progress concerning the studies of soil compressibility and creep properties via IL tests. It is generally believed that soil compressibility and creep during compression are relevant to its primary and secondary compression effect, including studies about the variation of $C_c$, $C_\alpha$, and $C_\alpha/C_c$. Specifically, Nash [6] found that the secondary compression index of over-consolidated soil tends to reach a constant value as time elapses. In early research, the variation trends of $C_\alpha$ with the change in vertical stress are different. Zhang et al. [7] found that the $C_\alpha$ of compacted loess under normal consolidation steadily decreased with increasing vertical stress. Wang et al. [8] studied the relationship between $C_\alpha$ and vertical stress and discovered that $C_\alpha$ tended to grow with the increase in vertical stress before stabilization. Ye et al. [9] studied the creep properties of compacted GMZ01 bentonite and found that $C_c$, $C_\alpha$, and $C_\alpha/C_c$ increased with the decrease in suction and increased with the increase in vertical stress. Ge et al. [10] studied and analyzed the effects of moisture content and vertical stress on the creep properties of compacted loess and found that $C_\alpha$ increased with increasing vertical stress. $C_\alpha$ was not sensitive to moisture content at a low level of stress, whereas it became more sensitive to moisture content at a high level of stress. Ajdari et al. [11] once examined the creep properties of the sand–bentonite mixture and found that $C_\alpha$ gradually decreased with an increase in suction. Wang et al. [12] investigated the deformation properties of two modified soils used as roadbeds under long-term loading and found that soil creep deformation and $C_\alpha$ increased with soil moisture content. Moreover, as vertical stress increased, the degradation of $C_\alpha$ accelerated. Usually, only one test method was employed for the research above; other test methods need to be simultaneously used to confirm the compressibility and creep properties.

Additionally, the constant rate of strain (CRS) method has been applied to study the compressibility and creep properties of soils. Sheahan and Watters [13] conducted nine CRS compression tests and four IL tests on Boston clay. They found that the compression parameters obtained using the two methods were in excellent agreement and concluded that the less structured clay might have a more negligible rate dependence during compression than the intergranular bounding and structured soil. Yin and Tong [14] found that bentonite–sand mixtures exhibited significant nonlinear swelling creep behavior based on IL and CRS tests, which indicated that the time-dependent properties of the buffer/backfill material were not negligible. Qin et al. [15] investigated the strain rate-related compression behavior of highly compacted GMZ01 bentonite. The results of single-stage and stepwise CRS tests were compared for a given suction to verify the applicability of the isotache concept of compacted GMZ01 bentonite. Díaz-Rodríguez et al. [16] found that the compressibility of Mexico City soils decreases with decreasing strain rate. The behavior of Mexico City soils during one-dimensional compression is strongly influenced by strain rate. Li et al. [17] investigated the effect of loading rate and its variation on the creep properties of mudstone and found that there was a significant loading rate effect in mudstone.

Many studies have been performed on the electrical resistivity properties of soil, which have reported that the resistivity decreases with the increase in vertical stress. In addition, the resistivity also decreases with increasing moisture content. Zhu and Li [18] developed a combined soil consolidation apparatus based on the principle of resistivity to research the soil at Lvliang Airport. They found that the resistivity of soil samples first decreased and then stabilized after applying loading. Dong et al. [19] studied the resistivity properties of loess under different stress, strain, porosity, saturation, and current frequency conditions. They found that the resistivity decreased with the increase in stress and saturation, the strain eventually stabilized when the loading was applied, and the resistivity also stabilized.



Badrzadeh et al. [20] measured the resistivity of saturated soils during consolidation and found that the resistivity is closely related to their physical properties. Qin et al. [21] studied the influence factors such as strain and current frequency on the resistivity of compacted loess under constant initial moisture content conditions. Duan et al. [22] studied the effect of moisture content on the resistivity of loess using an LCR digital bridge and found that the resistivity of loess decreased significantly with increasing moisture content. Therefore, the resistivity method is used in this work.

In some researches, soil microstructures were studied through the use of NMR and SEM. Li et al. [23] used quantitative and qualitative analysis of the microstructural characteristics of intact and remolded loess in different creep stages through SEM tests and found that the soil skeleton continuously changed during creep; the pore sizes and particle shapes had the most obvious variation. Casini et al. [24] used environmental scanning electron microscopy (ESEM) to study the arrangement of unsaturated silt particles and pores and found that with the increase in moisture content, the particle sizes increased while the volume of macropores decreased. In addition, the soil microstructure became denser after compression. Zheng et al. [25] conducted NMR tests on saturated remolded cohesive soil at different loading stages and qualitatively analyzed the relationship between transverse relaxation time $T_2$, peak area and porosity. Sun et al. [26] studied shale's pore structure and fractal characteristics using SEM and NMR. They found that the pore distribution of shale is irregular and complex, presenting multifractal and anisotropic characteristics.

This study investigates the primary compression index, secondary compression index and their ratio obtained when adopting two different loading methods (i.e., IL and CRS methods). In addition, the electrical resistivity during the compression of compacted loess was monitored and the microstructure of compacted loess was derived through the use of NMR and SEM techniques. Finally, the compressibility and creep properties, the electrical resistivity characteristics and the microscopical structures of the compacted samples were comprehensively compared and analyzed. This investigation can provide a reasonable reference to determine the compressibility and creep properties of loess.

## 2. Material and Methodology

### 2.1. Sample Preparation

2.1.1. Physical Properties of Loess

In this study, $Q_4$ loess was extracted in Dongshan, Taiyuan. The soil in this region is widely used as subgrade and ground-filling material. Importantly, the Dongshan region is a key protection region for landslides and geologic hazards in Shanxi Province. The depth of soil extraction is 1.2 m below the ground surface. After collecting from the site, the soil material was sealed in bags to avoid mixing impurities during transportation. The soil was homogeneous, yellow and hard; the mineral contents of the soil materials were determined through the use of X-ray diffraction: quartz (34.6%), sodium feldspar (19.4%), calcite (16.2%), muscovite (19.6%) and chlorite (10.2%). The physical properties of the soil samples are shown in Table 1, where proctor compaction tests determined the maximum dry density and optimum moisture content. The particle distribution curves are shown in Figure 1.

**Table 1.** Physical properties of loess.

| Specific Gravity $G_s$ | Maximum Dry Density $\rho_{max}$ (g·cm$^{-3}$) | Optimum Moisture Content $\omega_{opt}$ (%) | Liquid Limit $\omega_L$ (%) | Plastic Limit $\omega_P$ (%) | Plasticity Index $I_P$ | Collapsibility Coefficient $\Delta s$ |
|---|---|---|---|---|---|---|
| 2.77 | 1.72 | 15.3 | 22.8 | 13.1 | 9.71 | 0.02 |

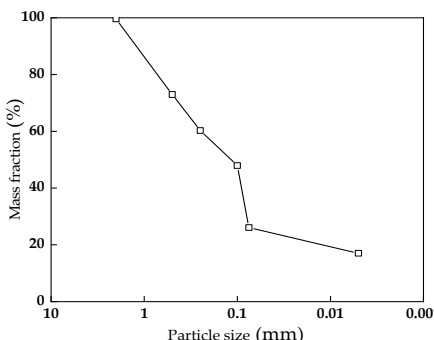

**Figure 1.** Particle distribution curve of the loess material.

### 2.1.2. Soil Sample Preparation

The collected loess was dried, crushed and passed through a 2 mm standard sieve to prepare the soil material. First, the soil material with an initial moisture content of 15% was prepared by spraying water and storing it in plastic bags for 24 h until the moisture was uniformly distributed in the soil. Then, the soil material was statically compacted at a constant rate of 0.4 mm/min to produce compacted samples with a diameter of 80 mm, a height of 10 mm and a dry density of 1.7 g/cm$^3$. In total, there are 16 samples in this study; 11 samples are used for the IL tests (the loading duration of every loading stage for samples N5-1, N10-1, N15-1, N20, A-1 and A-2 is a day, while samples N5-2, N10-2, N15-2, B-1 and B-2 is an hour. Samples A-1, A-2, B-1 and B-2 are used for microstructural tests after compression); and 5 samples are used for the CRS tests (samples H-5, H10 and H15 are only used for compression tests, while samples C-1 and C-2 are used for microstructural tests after CRS compression). The parameters of compacted samples are shown in Table 2.

**Table 2.** Table of parameters of compacted soil samples.

| Loading Method | Number | Water Content $\omega$ | Initial Dry Density $\rho_d$ (g·cm$^{-3}$) | Initial Pore Ratio $e_o$ | Initial Height of Soil Sample (mm) |
|---|---|---|---|---|---|
| IL | N5-1 | 5% | 1.698 | 0.5901 | 10.01 |
| | N10-1 | 10% | 1.675 | 0.6119 | 10.15 |
| | N15-1 | 15% | 1.667 | 0.6197 | 10.20 |
| | N5-2 | 5% | 1.693 | 0.5948 | 10.04 |
| | N10-2 | 10% | 1.693 | 0.5948 | 10.04 |
| | N15-2 | 15% | 1.698 | 0.5901 | 10.01 |
| | N20 | 20% | 1.667 | 0.6197 | 10.20 |
| | A-1 | 15% | 1.710 | 0.5789 | 9.940 |
| | A-2 | 15% | 1.698 | 0.5901 | 10.01 |
| | B-1 | 15% | 1.695 | 0.5929 | 10.03 |
| | B-2 | 15% | 1.690 | 0.5976 | 10.06 |
| CRS | H5 | 5% | 1.682 | 0.6052 | 10.11 |
| | H10 | 10% | 1.670 | 0.6168 | 10.06 |
| | H15 | 15% | 1.703 | 0.5854 | 9.980 |
| | C-1 | 15% | 1.702 | 0.5864 | 9.988 |
| | C-2 | 15% | 1.687 | 0.6008 | 10.08 |

### 2.2. *Test Apparatus*

The oedometer cell and the LCR digital bridge used in this study are shown in Figure 2. The oedometer cell is mainly composed of cotton yarn, a plastic bag, porous stones, an insulation ring, a copper electrode, a top cap and a bottom base. The sample is sandwiched between the upper and lower porous stones during the tests. In addition, filter paper is closely adjacent to the porous stone on the outside of the sample, which is not drawn in Figure 2. The horizontal resistivity of the sample was measured during the test using a pair of vertical copper electrodes with a width of 5 mm that was radially and symmetrically

adhered to the inner wall of the sample ring and connected to the resistivity meter through two conductive wires. The oedometer cell shown in Figure 2 was mounted on the IL or CRS loading frames for the corresponding compression tests, respectively.

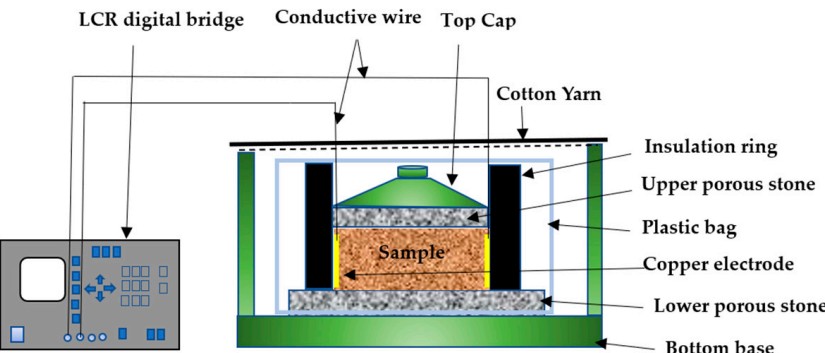

**Figure 2.** Schematic diagram of the oedometer cell and the LCR digital bridge.

The IL frame is a single-lever compression apparatus, which can be used for multistage loading compression tests, deriving the relationship between soil stress and deformation and calculating the compression index, rebound index and consolidation coefficient of soil. The loading frame has a loading stress range of 12.5–1200 kPa, a lever ratio of 12:1 and a cross-section area of 50 cm$^2$.

The CRS loading frame is a computer-controlled servo loading testing machine that has the advantages of a simple structure and a friendly operation, etc. It can be employed to conduct continuous loading compression tests. Its main technical parameters are as follows: the maximum test force of 50 kN, the relative error of the test force ±1%, the relative error of the beam displacement ±0.5%, the beam speed range of 0.0001–500 mm/min, the maximum travelling distance of the beam is 1100 mm.

The digital electric bridge used to monitor the electrical resistivity was the Tong-Hui TH2828ALCR digital bridge, which is automatically balanced. The instrument has a current frequency range of 50 Hz–1 MHz and an accuracy of 0.1%. The sample ring is made of insulated hard plastic, which has high strength and stiffness and can play an insulating role in preventing current loss. According to Dong et al. [19], the impedance corresponding to a current frequency of 50 kHz is collected in this paper, and the resistivity can be computed from the impedance using the following formula:

$$\rho = |Z| \cdot S/L \tag{1}$$

where $\rho$ is the resistivity, $\Omega \cdot m$; $|Z|$ is the impedance mode, $\Omega$; S is the area of the electrode, m$^2$; and L is the distance between the two electrodes, m.

During the compression tests, the measured deformation includes not only the deformation of the soil sample but also the deformation of the test apparatus. Therefore, in order to obtain only the deformation of the soil sample, it is necessary to perform displacement calibration of the IL and CRS loading tests to eliminate the deformation of the experimental apparatus. Figure 3 shows the deformation calibration curves of the experimental apparatus. As shown in Figure 3, the deformation of the apparatus is small when the vertical stress is slight, and the deformation of the experimental apparatus increases with the increase in vertical stress.

## 2.3. Test Scheme

The compacted soil sample was placed on the incremental loading frame, as shown in Figure 2. Then, an initial stress of 5 kPa was applied to ensure sufficient contact between the instrument and the sample. After the samples were dried or soaked to the designated moisture content, the incremental loading was started. During the loading process, distilled water is evenly sprayed onto the cotton yarn with a spray bottle at regular intervals to avoid

the effect of evaporation on the soil sample. According to the Standard for Geotechnical Test Methods (ASTMs D2435/D2435M-11(2020)), the loading steps were 25 kPa, 50 kPa, 100 kPa, 200 kPa, 400 kPa, 800 kPa and 1200 kPa; the stabilization standard was a deformation rate of no more than 0.005 mm/d. A digital micrometer was used to obtain the sample deformation at a specified time. Meanwhile, an LCR digital bridge was used to monitor the resistivity change in terms of the soil sample after loading.

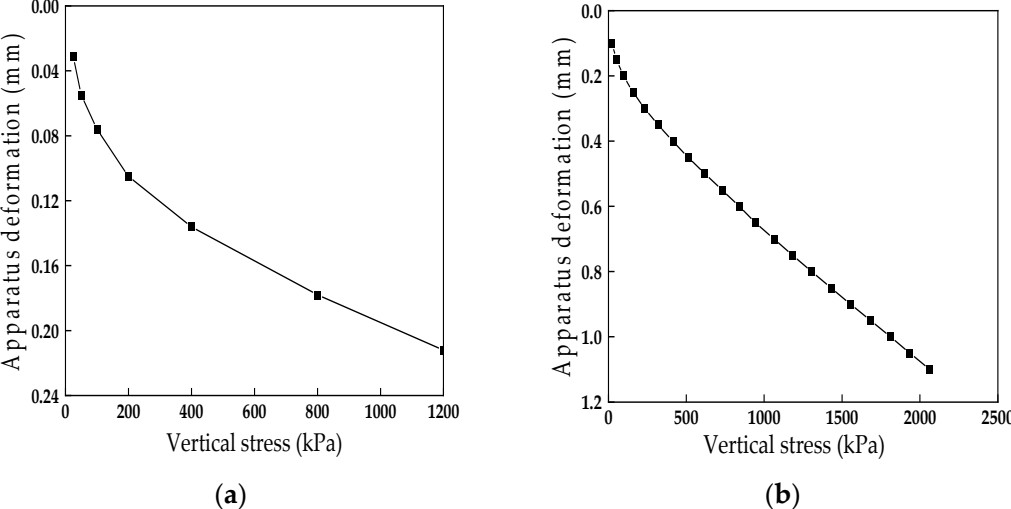

**Figure 3.** Deformation calibration curves of the test apparatus: (**a**) IL test apparatus; (**b**) CRS test apparatus.

The oedometer cell is mounted on the CRS loading frame for the constant rate of strain test, as shown in Figure 2. The strain rate in the CRS tests was conducted with reference to Qin et al. [15]. A stress of 5 kPa was applied to ensure the close contact of the test device components, and then the samples were compressed at a displacement rate of 0.06 mm/min for 1.8 mm, 0.006 mm/min for 0.6 mm, 0.0006 mm/min for 0.24 mm, and 0.06 mm/min for 0.6 mm, successively. The resistivity of the soil samples was monitored during the compression process.

After the compression, the microstructure of six samples with different loading processes was investigated using NMR and SEM, respectively. The loading processes of the samples are shown in Table 3. For NMR tests, the samples with R = 80 mm and H = 10 mm were first tested using the porous media fluid–structure coupling analyzer (MacroMR12-150H-I) to qualitatively analyze the relationship between the transverse relaxation time $T_2$, the spectral area and the void ratio. The NMR test was performed with a sampling frequency of 200 kHz, a main frequency of 12 MHz, a frequency bias of 771,645.95 Hz, an RF delay of 0.002 ms, a 90-degree pulse width of 9.00 μs, a 180-degree pulse width of 17.52 μs, a waiting time of 6000 ms, a return time of 0.5 ms and a peak offset of 0.02 ms. After the NMR tests, the central part of the specimen was taken to prepare clods with a size of $1 \text{ cm}^2 \times 0.5$ mm. After freezing the samples with liquid nitrogen, the samples were dried using a vacuum freezing dryer to better preserve the soil structure. Finally, Zeiss Gemini SEM 300 was used to take microscopic photographs of the soil clods. The operation procedures are as follows: (1) the treated clod was adhered to the platform with conductive adhesive, and then cladding material was sprayed on the surface of the clod; (2) the platform is put into the observation chamber for scanning, a magnification of 200 times was selected and the scanning images were saved. The procedure used to conduct the microstructural tests is shown in Figure 4.

**Table 3.** Loading processes of soil samples used for microstructural tests.

| Loading Method | Number | Loading Steps | Duration of Every Loading Stage |
|---|---|---|---|
| IL | A-1 | 25 kPa→50 kPa→100 kPa→200 kPa →400 kPa→800 kPa→1200 kPa | 1 day |
|  | A-2 | 25 kPa→50 kPa→100 kPa→200 kPa |  |
|  | B-1 | 25 kPa→50 kPa→100 kPa→200 kPa →400 kPa→800 kPa→1200 kPa | 1 h |
|  | B-2 | 25 kPa→50 kPa→100 kPa→200 kPa |  |
| CRS | C-1 | 0.06 mm/min—1200 kPa | —— |
|  | C-2 | 0.06 mm/min—200 kPa |  |

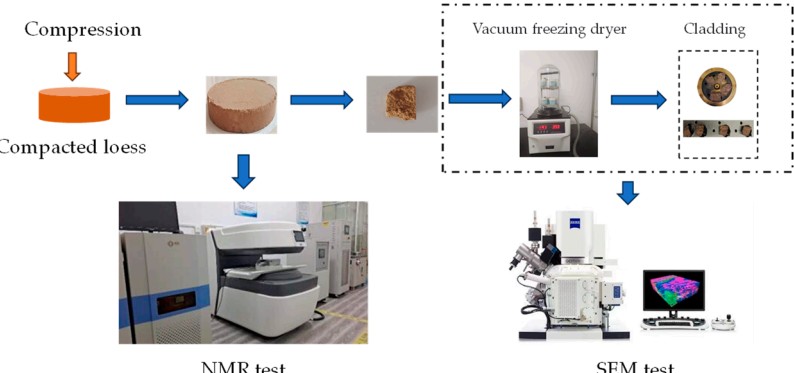

**Figure 4.** Process used in the microstructural tests.

## 3. Test Results and Analysis

### 3.1. Incremental Loading Tests

3.1.1. Compression Deformation

The relationship curves of the void ratio e and time t in semi-logarithmic coordinates for soil samples with different moisture contents under different vertical stresses were obtained via IL tests, as shown in Figure 5.

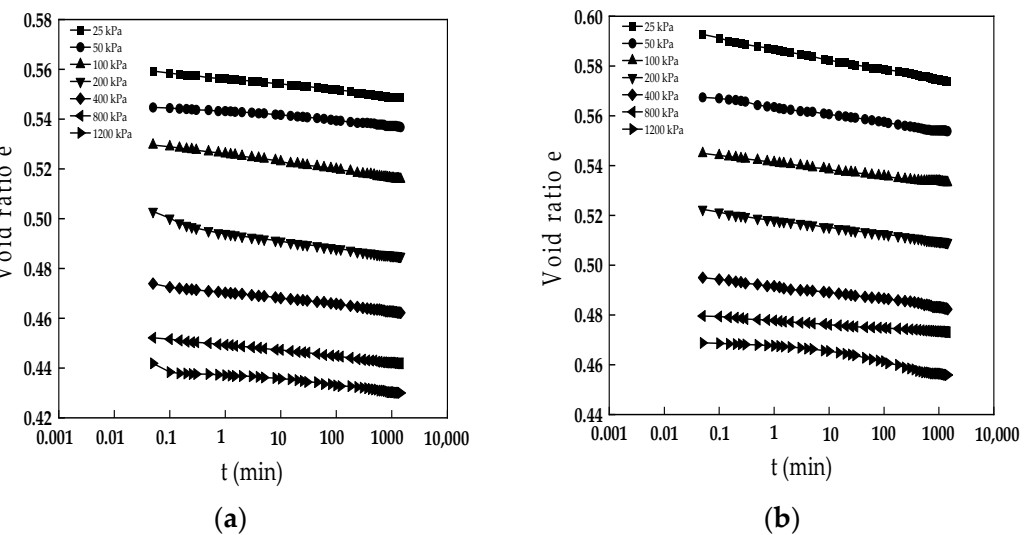

(**a**)                    (**b**)

**Figure 5.** *Cont.*

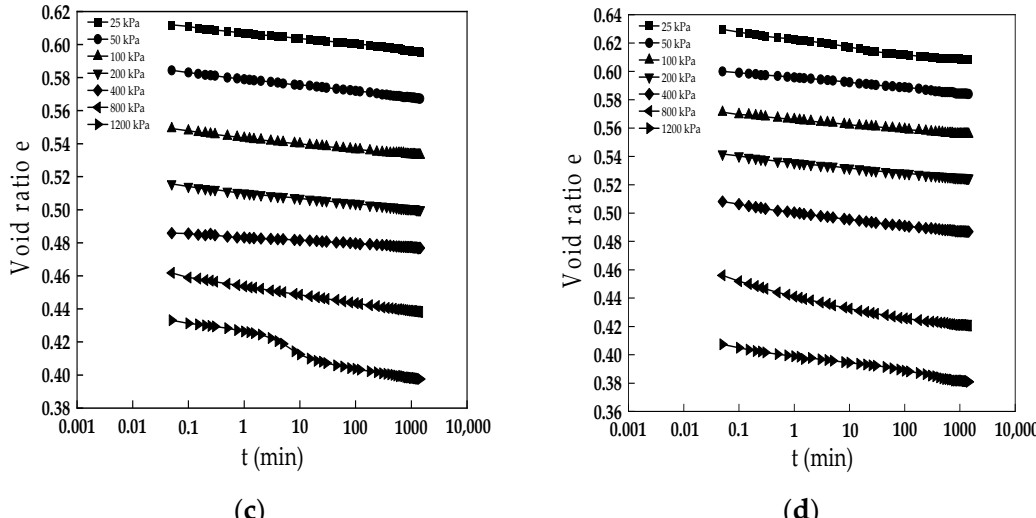

(**c**)                                                                                    (**d**)

**Figure 5.** Curves of void ratio versus logarithmic time: (**a**) N5-1; (**b**) N10-1; (**c**) N15-1; (**d**) N20.

As shown in Figure 5, the void ratio of the soil samples gradually decreases as vertical stress increases, and the rate of void ratio decreases progressively and slowly with the increased loading time. The e-logt curves present linear evolution when the vertical stress is small. However, the curves showed an inverse S-shape when the vertical stress is large. Therefore, it is not easy to directly distinguish between primary and secondary compression when using e-logt curves. According to Chen [27], this paper divides soil samples' primary and secondary compression demarcation points by the vertical deformation rate–vertical deformation curve. As shown in Figure 6, the vertical deformation rate $S_t'$-vertical deformation $S_t$ curves at various vertical stress levels can be divided into three segments. The first segment presents a significantly decreasing vertical compression deformation rate, which is the main compression part. The second segment is the middle inflection part, which is the transition stage of primary and secondary compression. The third segment is the stable stage, and the rate closes to zero, which is the second compression stage.

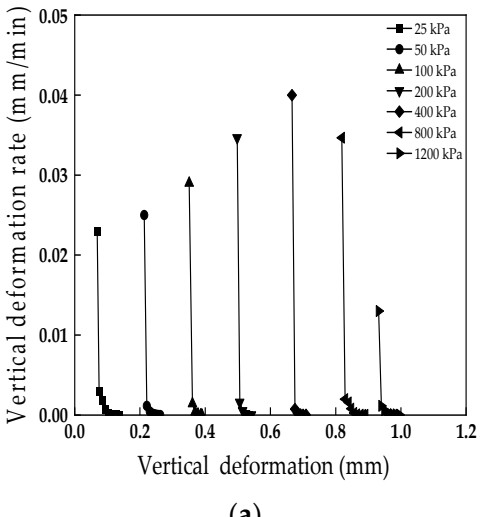
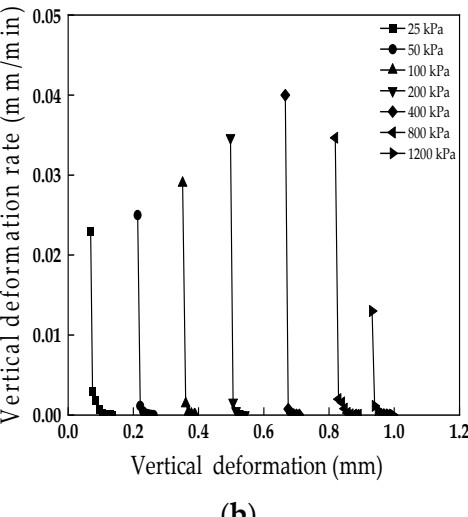

(**a**)                                                                                    (**b**)

**Figure 6.** *Cont.*

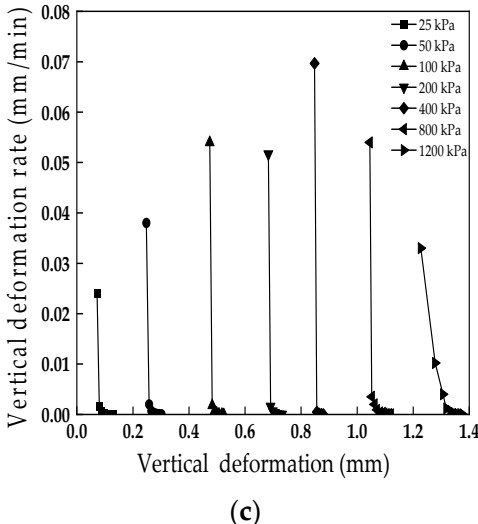
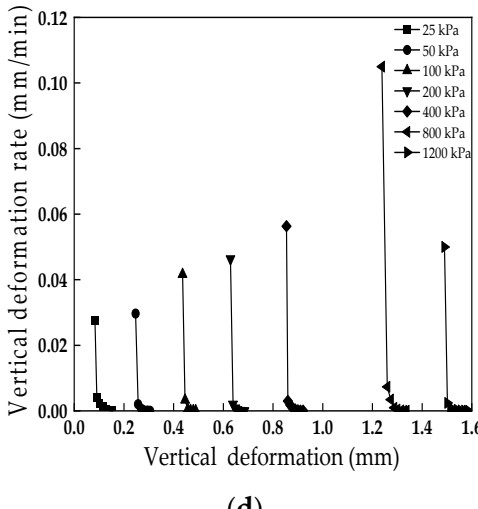

**(c)**                                      **(d)**

**Figure 6.** Vertical deformation rate versus vertical deformation: (**a**) N5-1, (**b**) N10-1, (**c**) N15-1, (**d**) N20.

The secondary compression index can be determined by using the slope of the secondary compression part of the e-logt curve. The formula for calculating the secondary compression index is shown in Equation (2) [9]:

$$C_\alpha = \Delta e / (\mathrm{log} t_2 - \mathrm{log} t_1) \tag{2}$$

where $t_1$ is the moment of completion of primary compression, min; $t_2$ is the moment of the end of compression, min; $\Delta e$ is the difference of the void ratio in the secondary compression part.

Figure 7 shows the relationship between $C_\alpha$ and vertical stress σ. It can be seen that the $C_\alpha$ values of unsaturated compacted loess show an increasing trend with the increase in vertical stress. The larger the moisture content, the more noticeable this trend because the creep and particle rearrangement and the pore collapse rate increase as the vertical stress increases, which augments the macroscopical expression of the secondary compression index [10].

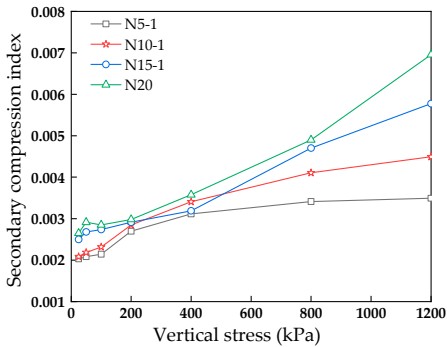

**Figure 7.** Curves of secondary compression index versus vertical stress.

As the moisture content is high at a given density, there is more free water between soil particles; therefore, the adsorbed water film thickness of the soil is thicker, and the capillary suction between soil particles is smaller, which makes the soil more susceptible to slip and creep, which macroscopically manifests as an increase in the secondary compression index [28].

$C_c$ and $C_\alpha$ reflect the compression and creep properties of the soil. $C_\alpha$ reflects the secondary compression of the soil, while $C_c$ reflects the primary compression of the soil. In order to determine the compression properties corresponding to every loading step, the

incremental primary compression index $C_c^*$ is defined in Equation (3) [9]. The formula for $C_c^*$ is shown below:

$$C_c^* = \Delta e / \Delta \log \sigma^*$$ (3)

where $\Delta e$ is the void ratio difference value corresponding to a loading increment, $\Delta \log \sigma^*$ is the difference value of logarithmic stress corresponding to the void ratio increment.

The normalized void ratio $e/e_o$ shown in Figure 8 was used to eliminate the effect of the initial void ratio. It can be observed in Figure 8 that $e/e_o$ decreases with the increase in vertical stress at a certain moisture content. In addition, $e/e_o$ decreases with the increase in moisture content for a given vertical stress, which shows that the compressibility of the soil sample increases as the initial moisture content increases.

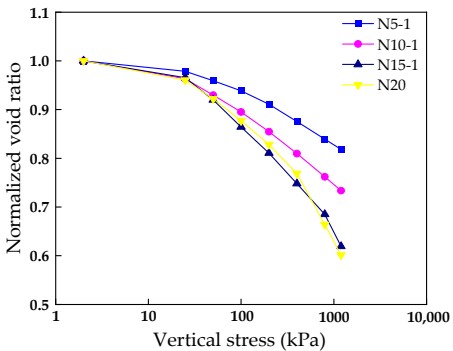

**Figure 8.** Curves of normalized void–logarithmic stress of samples with different moisture content.

It is essential to note that $C_c^*$ is different from $C_c$, which refers to the standard compression slope of the curve [9]. According to Equation (3), the $C_c^*$ of the soil samples were obtained under different vertical stresses. Figure 9 shows the curves of $C_c^*$ and $\sigma^*$. It can be observed that $C_c^*$ increases with the increase in moisture content for a given vertical stress and increases with the increase in vertical stress for a given moisture content. Ye et al. [9] and Hu et al. [29] found that the $C_c$ values of GMZ01 bentonite and loess were affected by the vertical stress and suction in the compression tests of unsaturated soil, and the $C_c$ values increased with the increase in vertical stress and with the decrease in suction. Previous studies found a close relationship between the suction of soil samples and their moisture content. The matrix suction of unsaturated loess gradually decreases with increasing moisture content [30]. It can be seen that the moisture content has a significant effect on the compression index of the soil samples.

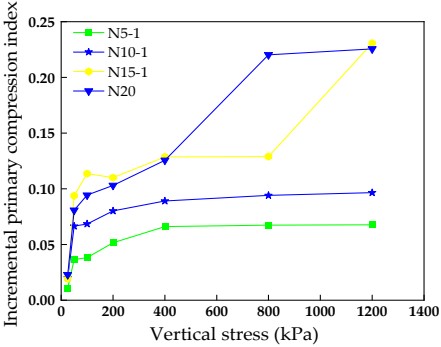

**Figure 9.** Curves of incremental primary compression index versus vertical stress.

Note that $\alpha = C_\alpha / C_c^*$ is an important parameter to evaluate secondary compression deformation. Figure 10 shows the relationship of $C_\alpha$ and $C_c^*$ under the effects of different moisture contents. The relationship between $C_\alpha$ and $C_c^*$ has been extensively studied; for example, Mesri and Godlewski [31] have found that $\alpha$ is a constant value for saturated intact soil. In Figure 10, it can be seen that the $\alpha$ values of the unsaturated samples were also

constant values, and their ranges were between 0.026 and 0.042. In addition, α increases with the increase in moisture content. Other researchers have found similar variations. Studying the secondary compression of unsaturated GMZ01 bentonite, Ye et al. [9] found that α existed in a narrow range and increased with decreasing suction.

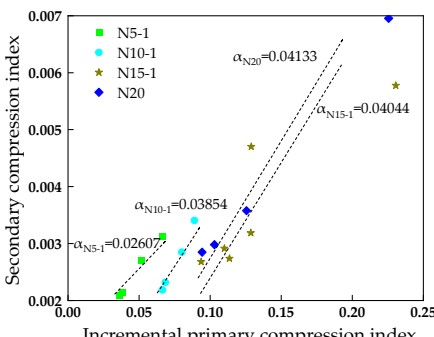

**Figure 10.** Relationship between secondary compression index and incremental primary compression index.

### 3.1.2. Resistivity Properties

The resistivity method has the advantages of convenience, continuity, speed, and economy. The resistivity properties can be monitored during compression [32,33].

Figure 11 shows the curves of resistivity–time obtained using the IL tests. As shown in Figure 11, it seems that the resistivity decreases with the increase in moisture content and vertical stress, but the resistivity of the soil sample first rapidly decreases, and then slightly increases as time increases for a given loading stage. It seems that the resistivity of samples depends on the connected channels filled by water. As the vertical stress increased, the channels that filled the water increased; thus, the resistivity decreased. However, due to water loss from the channels, the connected channels decreased, and the resistivity increased.

### 3.2. CRS Tests

The IL test equipment is simple, but it also has disadvantages like a time-consuming test period, discrete test data and human interference. On the contrary, the CRS method can record data continuously, and the test period is short, which can make up for the shortcomings of the IL method. In this paper, the creep properties of unsaturated compacted loess are also studied through the use of CRS tests.

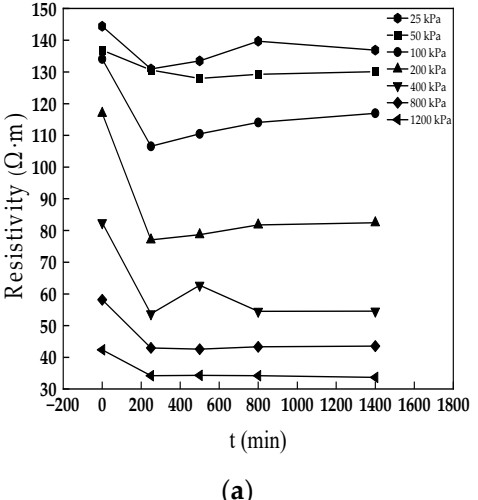

(**a**)

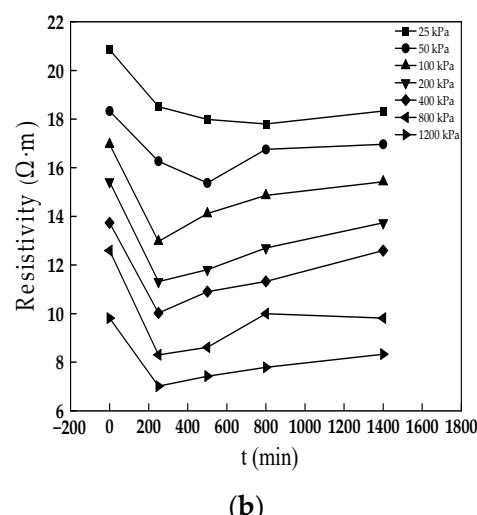

(**b**)

**Figure 11.** *Cont.*

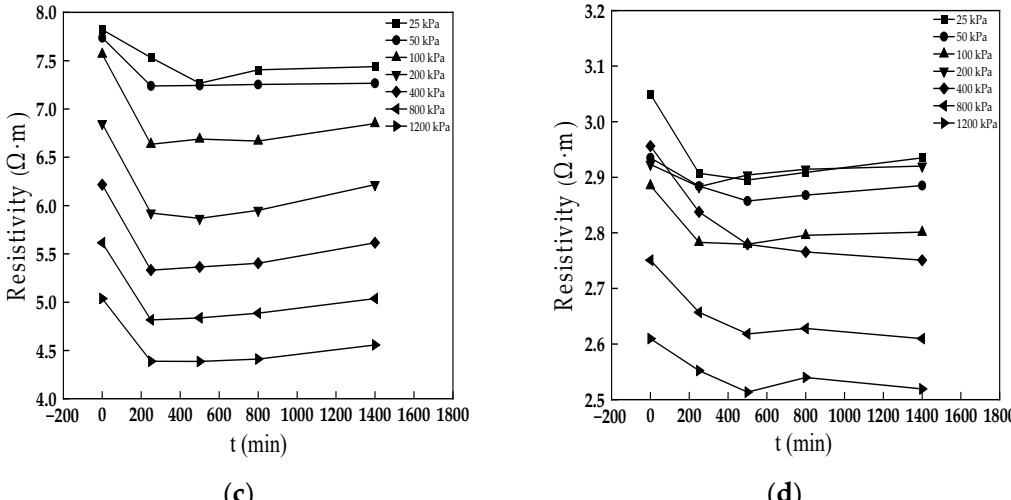

**Figure 11.** Curves of resistivity versus time: (**a**) N5-1, (**b**) N10-1, (**c**) N15-1, (**d**) N20.

### 3.2.1. Compression Deformation

Figure 12 shows the stress–strain curves obtained via the CRS tests, which shows that the strain ε increases with the vertical stress σ. When the strain rate changes, there is a jump that happens in the ε-σ curve. The strain increases slowly when the vertical stress is small, which is the elastic stage. When the vertical stress is large, the strain obviously increases, which is the plastic stage. In other words, the soil sample undergoes both elastic and plastic deformation. In addition, the vertical strain increases for a given vertical stress as the moisture content increases.

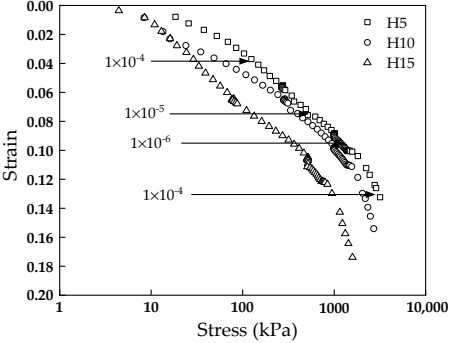

**Figure 12.** Stress–strain curves of the CRS tests.

Figure 13a shows the curves of the void ratio e and logarithmic vertical stress logσ, and the compression index $C_c$ can be determined in terms of the e-logσ curve, which is 0.1986 for H5, 0.2901 for H10 and 0.3796 for H15. The $C_c$ values obtained when using the CRS and the IL tests have the same variation trend. Namely, $C_c$ increases with the increase in moisture content. Figure 13b shows that the void ratio of the soil samples gradually decreases with the increase in stress after applying loading with different strain rates to the soil samples. At the beginning of loading, the soil is loose, and the loading rate is fast; therefore, the vertical stress of the soil sample increases slowly with the decrease in void ratio. As loading continues, the void ratio decreases continuously because the pore volume gradually decreases. The void ratio changes at a reduced rate because the soil mass is gradually compacted, and its resistance to deformation increases, as shown in Figure 13b.

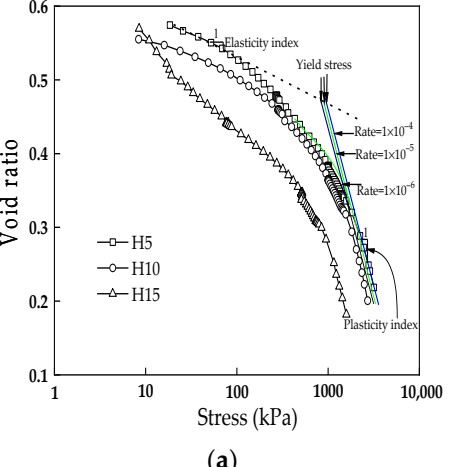
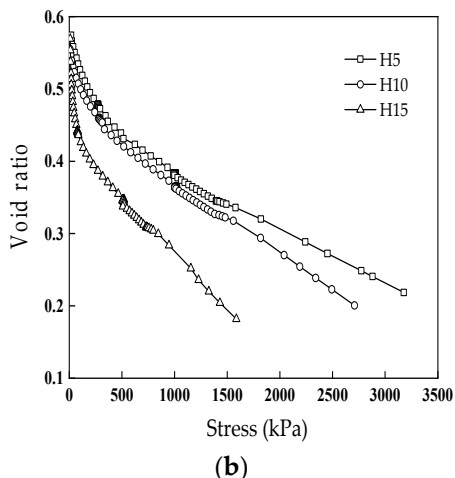

**Figure 13.** Compression curves of the CRS tests. (**a**) curves of void ratio–logarithmic stress; (**b**) curves of void ratio–stress.

It was shown that there is a relationship between the secondary and the primary compression index [31]. Gennaro and Pereira [34] reported that there was a relationship between the yield stress $\sigma_y$ and the strain rate, as shown in Equation (4):

$$\log \sigma_y = A + \alpha \Delta \log \dot{\varepsilon} \tag{4}$$

where A and $\alpha$ are material parameters; $\sigma_y$ is the yield stress, kPa; and $\dot{\varepsilon}$ is the strain rate, s$^{-1}$.

Due to the fact that Equation (4) is differential, Equation (5) can be obtained as follows:

$$\Delta \log \sigma_y = \alpha \Delta \log \dot{\varepsilon} \tag{5}$$

According to Qin et al. [21], the yield stress is determined by analyzing the intersection of the tangent of the elastic and plastic segments of the compression curve. Namely, the stress corresponding to the intersection point is the yield stress; the yield stress depends on the strain rate, as shown in Figure 13. Figure 14 shows the curves of logarithmic yield stress versus logarithmic strain rate. The slope of the curve in Figure 14 can be obtained in terms of Equation (5) and is equal to the value of $\alpha$, which is the ratio of the secondary compression index to the primary compression index. According to Figure 14, it can be determined that $\alpha = 0.035$ for H5, $\alpha = 0.044$ for H10 and $\alpha = 0.055$ for H15. Based on the primary compression index obtained above, the secondary compression index of the soil sample can be calculated as $C_\alpha = 6.95 \times 10^{-3}$ for H5, $C_\alpha = 12.76 \times 10^{-3}$ for H10 and $C_\alpha = 20.88 \times 10^{-3}$ for H15. It can be seen that the creep parameters derived from the CRS and IL tests had the same variation trend. Both $\alpha$ and $C_\alpha$ increase with the increase in moisture content.

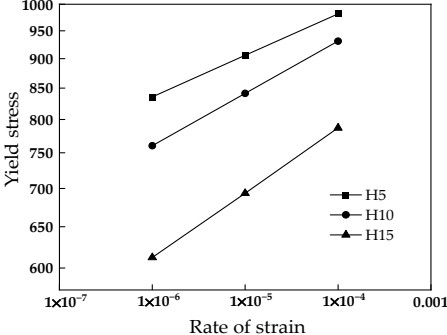

**Figure 14.** Relationship of logarithmic yield stress and logarithmic rate of strain.

### 3.2.2. Resistivity

Figure 15 shows the curves of the resistivity ρ-strain ε. It can be seen that the smaller the moisture content, the greater the resistivity, indicating that there are fewer connected channels [22,35,36]. As the soil sample is compressed, the resistivity of the soil sample tends to decrease. Furthermore, there were two segments compressed by the strain rate of $1 \times 10^{-4}$ s$^{-1}$; the resistivity of the former segment decreased more quickly than the later segment. Interestingly, when the samples were compressed with a minimum strain rate of $1 \times 10^{-6}$ s$^{-1}$, an increase in resistivity was observed. This may be due to the loss of free water during compression, reducing the connected channels of the soil sample.

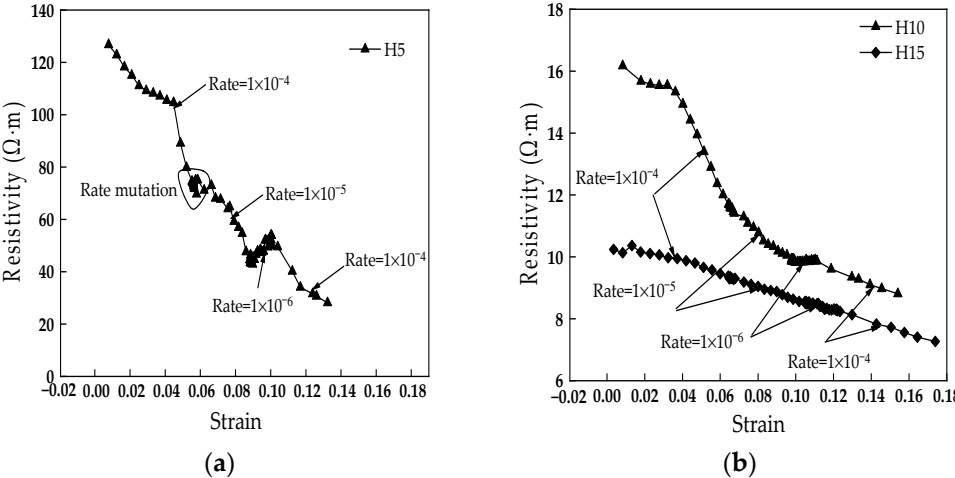

**Figure 15.** Curves of resistivity versus strain (**a**) resistivity curve of sample H5, (**b**) resistivity curve of samples H10 and H15.

### 3.3. Microstructural Tests

### 3.3.1. NMR Tests

Relaxation is an incredibly significant physical property in the context of NMR. The relaxation time serves as a time constant for the decay of the magnetization vector. There are two main relaxation processes: longitudinal relaxation time $T_1$ and transverse relaxation time $T_2$. In NMR, the transverse relaxation time $T_2$ is commonly employed to characterize the soil pore structure. $T_2$ is influenced by the proton relaxation time inside the pores $T_{2B}$ and the surface relaxation time $T_{2S}$ [37]. Equation (6) illustrates this relationship.

$$\frac{1}{T_2} = \frac{1}{T_{2B}} + \frac{1}{T_{2S}} = \frac{1}{T_{2B}} + \rho_2 \left(\frac{S}{V}\right) \text{pore} \tag{6}$$

$$\frac{1}{T_2} \approx \frac{1}{T_{2S}} = \rho_2 \left(\frac{S}{V}\right) \text{pore} \tag{7}$$

where $\rho_2$ is the surface relaxation intensity (μm/ms), and (S/V) pore is the specific surface area of pores (μm$^2$/μm$^3$).

Therefore, the $T_2$ distribution serves as an indicator of pore size distribution in the soil. The position enclosed by the spectral curve in the $T_2$ distribution curve corresponds to the pore size, while the area of the spectral peak correlates with the pore volume of the soil. Specifically, small $T_2$ values indicate small pore sizes. Correspondingly, the small area enclosed by the spectral curve means small pore volumes [25].

As shown in Figure 16, the $T_2$ distribution curves of the samples under different loading processes display three peaks: the peak with little relaxation time is peak 1 (1–5 ms), the peak with moderate relaxation time is peak 2 (30–200 ms), and the peak with a large relaxation time is peak 3 (500–1200 ms). Combined with the results of Zheng et al. [25], the comprehensive analysis suggests that peak 1, peak 2 and peak 3 correspond to micropore size, mesopore size and macropore size, respectively. The difference in these peaks in

terms of NMR signals is caused by the uneven particle grading of the soil samples [25]. In Figure 16, it can be seen that peak 2 and peak 3 shifted to the left with the increase in vertical stress, indicating that the pore diameter of the macropores of the soil samples gradually decreased during the compression process. Meanwhile, the areas enclosed by the spectral curve around peaks 2 and 3 also decreased with the increase in vertical stress, indicating that the free water in the macropores was gradually discharged during compression and the pore volume gradually decreased with compression. Compared with (a), (b) and (c) in Figure 16, the variation of peak 1 of the samples in Figure 16a is different from that in Figure 16b,c. In Figure 16b,c, the spectral curve area around peak 1 with low loading is larger than that with high loading, indicating that the volume of micropores increases with vertical stress during compression. However, the area enclosed by the spectral curve around peak 1 shows the opposite variation in Figure 16a. This may be attributed to more pore water loss from the soil sample during the longer loading duration. Since the NMR signal is closely related to the content of hydrogen-containing fluid (pore water) in the pores, the signal intensity of NMR decreases with the decrease in moisture content. Meanwhile, there is a certain difference between the pore space detected through the use of NMR and the actual pore space of the soil samples for the unsaturated soil. In order to further explain the variation in the NMR curves in Figure 16, the moisture content of the soil samples after loading for the NMR tests is shown in Figure 17. In Figure 17, it can be observed that the moisture content of the soil samples decreases with the increase in vertical stress, with an especially obvious decrease with increased loading duration.

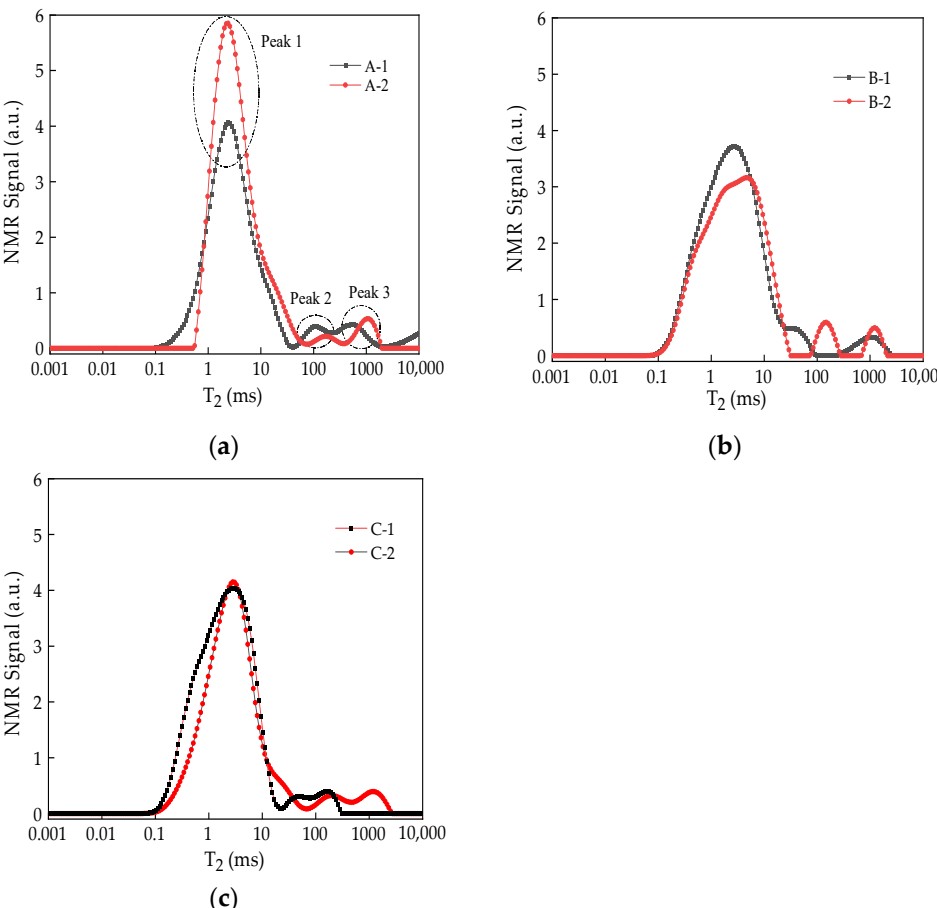

**Figure 16.** Curves of NMR spectra (**a**) NMR curves of samples A-1 and A-2, (**b**) NMR curves of samples B-1 and B-2, (**c**) NMR curves of samples C-1 and C-2.

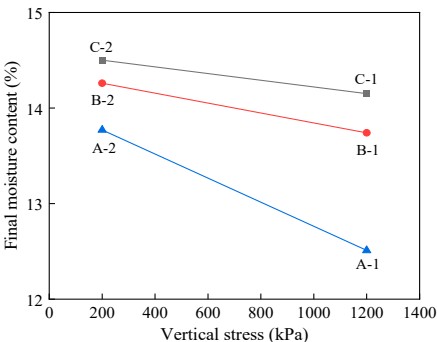

**Figure 17.** Moisture content of soil samples after loading.

3.3.2. SEM Tests

Four clods extracted from the samples under different loading processes were selected for SEM tests (A-1, A-2, C-1, C-2). Figure 18 shows the SEM images of the four clods at a magnification of 200 times. In this study, the microstructure of unsaturated compacted loess under different compression processes was qualitatively analyzed from the perspectives of pore state, particle size and particle contact relationship. Through comparison of the SEM images in Figure 18, it is found that the soil skeleton is looser when the vertical stress is slight and there are some macropores with pore sizes ranging from 4 um to 200 um. Meanwhile, the particle size of the soil clods was obviously coarse, and the particle morphology was mainly granular and blocky. In addition, the contact relationships between the particles were mostly point-to-point contacts and edge-to-edge contacts. Due to large macropore sizes and particles, as well as the unstable particle contact relationship, it is easy to compress the soil samples and achieve large electrical resistivity under slight vertical stress. With the increase in vertical stress, the volume of the pores of the soil sample decreases. Macroscopically, this is manifested by a decrease in the soil samples' vertical compression and resistivity. In addition, the coarse particles of the soil samples were broken, which led to an increase in the number of fine particles. The fine particles were embedded with each other to form stable particle agglomerates. The contact relationship between the particles also transitioned to a stable and metastable state dominated by face-to-face contacts after the implementation of large vertical stress. The increase in the number of fine particles and the transformation of the contact relationship caused significant compressive deformation under large vertical stress. Through comparison of the SEM images of the samples after the IL tests, those that underwent the CRS tests have denser soil skeletons, more homogeneous particle sizes and finer particles for a given vertical stress. It indicates that the volume change in the soil samples that underwent the CRS tests is larger than that underwent the IL tests for a given vertical stress. Correspondingly, the compression parameters obtained using the CRS tests are larger than those obtained using the IL tests.

In addition, the SEM images of unsaturated compacted loess were quantitatively analyzed in this study using Image-Pro Plus (IPP) 6.0 software. In order to distinctly observe the particles and pores, the images were binarized to white for the pores and black for the particles as shown in Figure 19. After analysis, the percentage of the pore area of the soil samples is 25.24% (A-1), 32.28% (A-2), 11.13% (C-1) and 15.98% (C-2), respectively, indicating that the pore area decreased as the vertical stress increased and the samples after CRS tests has a smaller pore area than those after the IL tests for a given vertical stress. Meanwhile, the shape change in loess particles was quantitatively analyzed by calculating two parameters. One of them is the abundance, C, which is defined as the ratio of the short diameter to the long diameter of the particle; the other is the roundness, R, which is defined as the ratio of the circumference of the equivalent area of the particle to the actual circumference of the particle projection. In light of the work of Zhang and Wang [38], the C value can roughly characterize the sharpness of loess particles, while R reflects the extent to which soil particles tend to be rounded. The equations used to determine C and R are

shown below. According to Equations (8) and (9), the C value ranges from 0 to 1, and the R value increases as the particle shape inclines to a circle.

$$C = \frac{B}{L} \tag{8}$$

$$R = \frac{2\sqrt{\pi A}}{P} \tag{9}$$

where L is the length of the long axis of loess particles, B is the length of the short axis of loess particles, A is the area of loess particles, and P is the circumference of the projected contour of the particle.

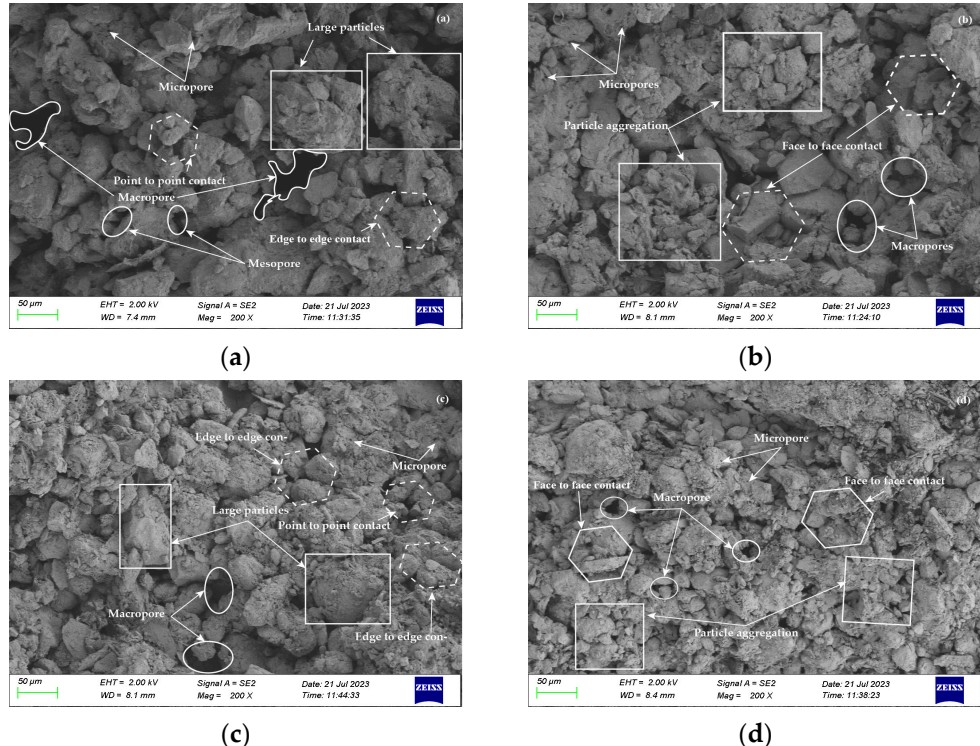

**Figure 18.** SEM images of unsaturated compacted loess (**a**) A-2, (**b**) A-1, (**c**) C-2, (**d**) C-1.

The abundance of the compacted samples is shown in Figure 20a. The abundance of samples ranging from 0.9 to 1 dominated in terms of the distribution, their percentages increasing as the vertical stress increased. According to Zhang and Wang [38], as the C value is closer to 1, the shape of the particle is inclined to be equiaxial in shape. Thus, the proportion of approximately equiaxial particles increases as the vertical stress increases due to the fact that the number of coarse particles being crushed to fine particles increases. The roundness distribution of the soil samples is shown in Figure 20b. The roundness value of compacted loess samples lies in the scope of 0.5–1.3 and is dominated by the range of 1.1–1.3. The proportion of roundness ranging from 1.1 to 1.3 increases as vertical stress increases, indicating that the roundness of particles increases as the soil particles are crushed. According to Fang et al. [39], the soil particles are closely arranged with large areas and small circumference as the vertical stress increases. Moreover, irregular particles are easier to compress than packed round particles [40]; the particles are gradually transformed from irregularly elongated to equixially rounded particles as the vertical stress increases. The pore area of the soil sample decreases with the increase in particle abundance and roundness as the large macropores are compressed into smaller macropores [41].

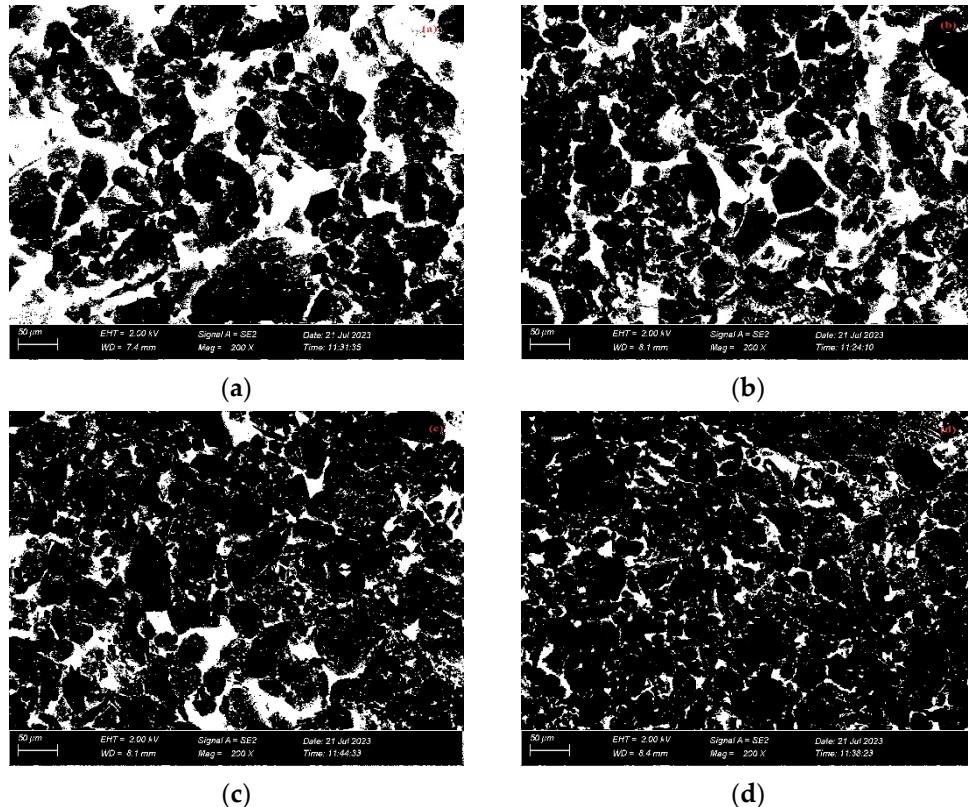

**Figure 19.** Binarized images (**a**) A-2, (**b**) A-1, (**c**) C-2, (**d**) C-1.

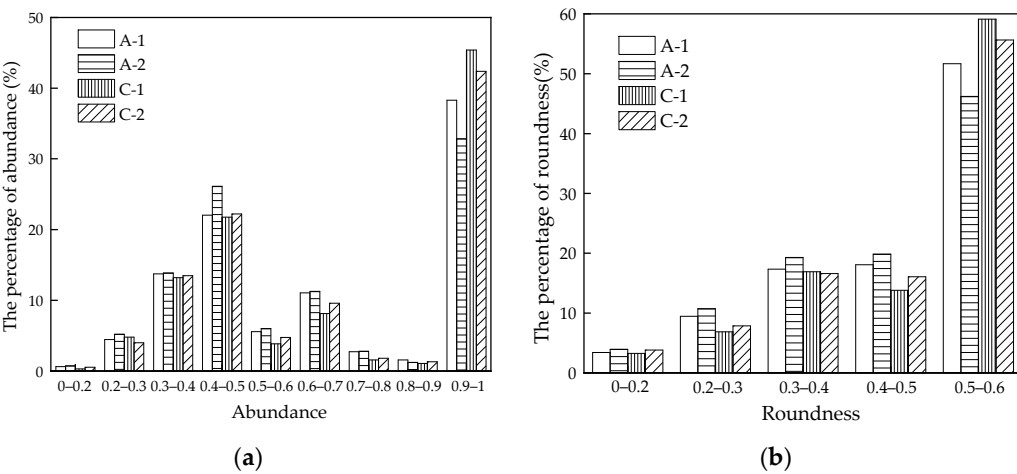

**Figure 20.** Distribution of particle abundance and roundness of compacted samples: (**a**) abundance distribution; (**b**) roundness distribution.

In addition, the abundance, ranging from 0.9 to 1, and the roundness, ranging from 1.1–1.3, of the samples after the CRS tests are larger than those after the IL tests for a given loading, which may be attributed to the influence of water. Compared with the samples after the IL tests, there is much more water reserved in the samples after the quick CRS tests. The irregularly elongated particles are more vulnerable in terms of being crushed to equixially rounded particles due to their high water content.

## 4. Discussion

### 4.1. Compression Properties

In this study, both $C_c$ and $C_\alpha$ increased with the increase in moisture content and vertical stress, which is similar to the results obtained by Ge et al. [10]. However, different

variations influenced by the vertical stress were also found. Wang et al. [8], Zhou and Chen [42] and Jiang [43] found that the secondary compression index increased with the increase in vertical stress and eventually converged to a stable value. Figure 21a shows the curves of secondary compression index–vertical stress reported by Wang et al. [7], which was attributed to the gradual completion of the damage and adjustment of the clay particles with the increase in vertical stress. Thus, the secondary compression deformation and index eventually tend to stabilize.

Yin et al. [44] and Yu et al. [45] found that the secondary compression index of soil samples first increases and then decreases with the increase in vertical stress. The curve of the secondary compression index–vertical stress reported by Yu et al. [45] is shown in Figure 21b. At the beginning of the over-consolidation state, the secondary compression index is slight due to low vertical stress. The secondary compression index gradually increases as the vertical stress increases close to normal consolidation. In a normal consolidation state, the secondary compression index decreases as the vertical stress increases due to soil compaction [45].

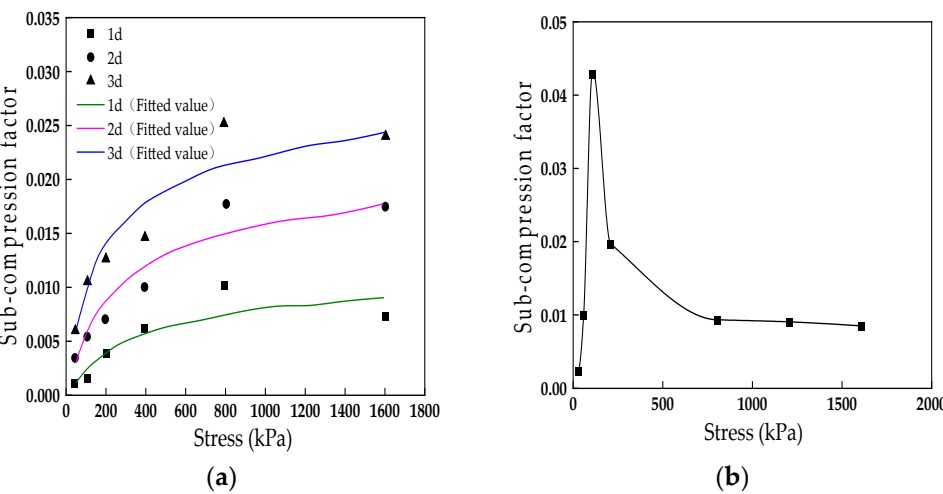

(a)                                        (b)

**Figure 21.** The curves of secondary compression index–vertical stress: (**a**) Wang et al. [8]; (**b**) Yu et al. [45].

### *4.2. Comparison of IL and CRS Tests*

#### 4.2.1. Compressibility

The device of the IL test is simple, but it is time-consuming and influenced by human interference. The CRS test can continuously collect test data and has a shorter test period, which can make up for the shortcomings of the IL test. Figure 22 shows the curves of the normalized void ratio versus logarithmic stress obtained when using the IL and CRS tests. The variation in compression deformation obtained via the two tests is highly similar. The normalized void ratio obtained via the CRS test is less than that obtained when using the IL test for a given vertical stress and moisture content. The figure shows that the primary compression index increases with increasing moisture content. In addition, the primary compression index obtained through the use of the CRS test is greater than that obtained via the IL test.

Some comparisons of the CRS and IL tests were performed. Nguyen et al. [46] analyzed the consolidation properties of soft clay in Vietnam by combining the CRS and IL tests and found a high similarity between the test results obtained via the two test methods. In Figure 23a, Andries et al. [47] summarized and compared the IL and CRS results of two intact over-consolidated clay samples. They found that the compressibility obtained when using CRS tests and IL1 tests was very similar, which was smaller than the curve obtained via the IL2 test. In addition, Leroueil [48] found that the curve of the void ratio versus the vertical stress of saturated soil obtained when using CRS tests is above that obtained via the IL test because the strain rate is large for CRS compression.

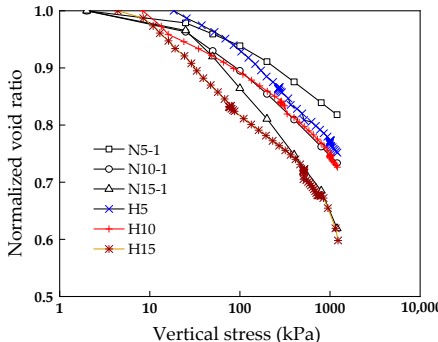

**Figure 22.** Curves of normalized void–logarithmic stress of samples with different moisture content and loading duration.

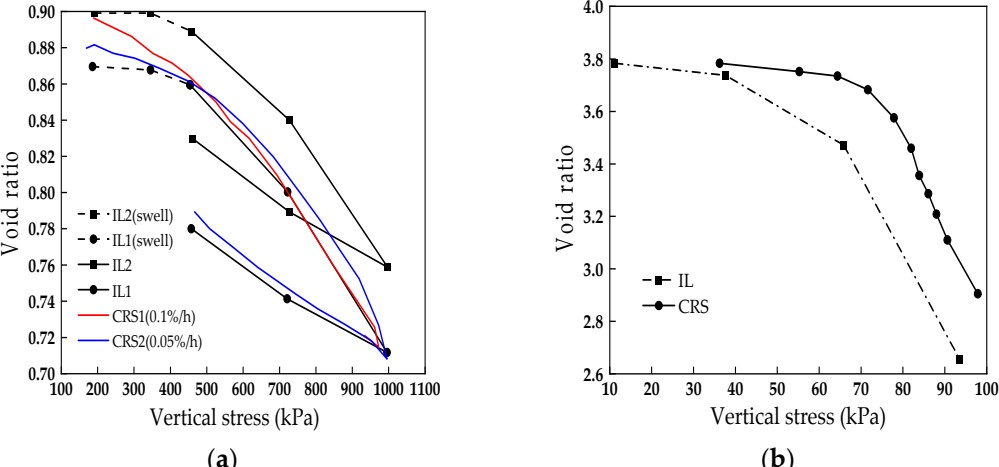

(**a**)            (**b**)

**Figure 23.** Comparison curves of void ratio versus vertical stress: (**a**) Andries et al. [47]; (**b**) Leroueil [48].

In addition, there have been studies that used the CRS and IL methods for the analysis of unsaturated soil. In Figure 24, a comparison of the compression curves obtained when using the IL and CRS tests reported by Cui and Delage [49] is shown. It is observed that the compressibility of the soil sample decreases with the increase in suction. Since the suction increases with a decrease in moisture content, the compression properties influenced by moisture content in this work have the same variation trends as the results in Cui and Delage [49]. In addition, it seems that the compression curves obtained when using CRS and IL tests have little difference in terms of plastic compression, which is a little different from the variation noted in the results of this work. This may be attributed to the differences in terms of controlling suction and initial moisture content.

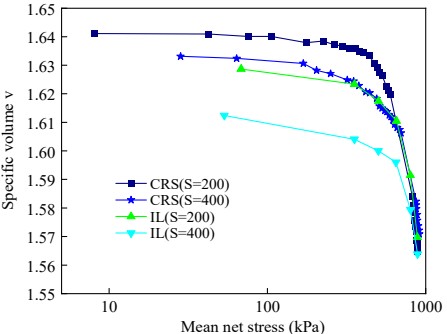

**Figure 24.** Compression curves obtained when using the IL and CRS tests reported by Cui and Delage [49].

In this study, two IL test schemes were conducted to research the influence of loading duration on the deformation of samples. The two schemes have identical loading steps: 25 kPa, 50 kPa, 100 kPa, 200 kPa, 400 kPa, 800 kPa and 1200 kPa. However, one scheme used each loading step for a day, while the other applied each step for an hour. This is because the end time of the primary compression of the soil sample is about an hour. Figure 25 shows the comparison of the IL tests with different loading durations. It can be seen that the compression deformation increment of the soil samples undergoing loading for an hour is larger than that for a day for a given initial moisture content, which was attributed to the variation in moisture content with loading duration. In other words, the shorter the loading duration, the larger the moisture content, indicating that compression with a short loading duration discharges less water in the IL tests.

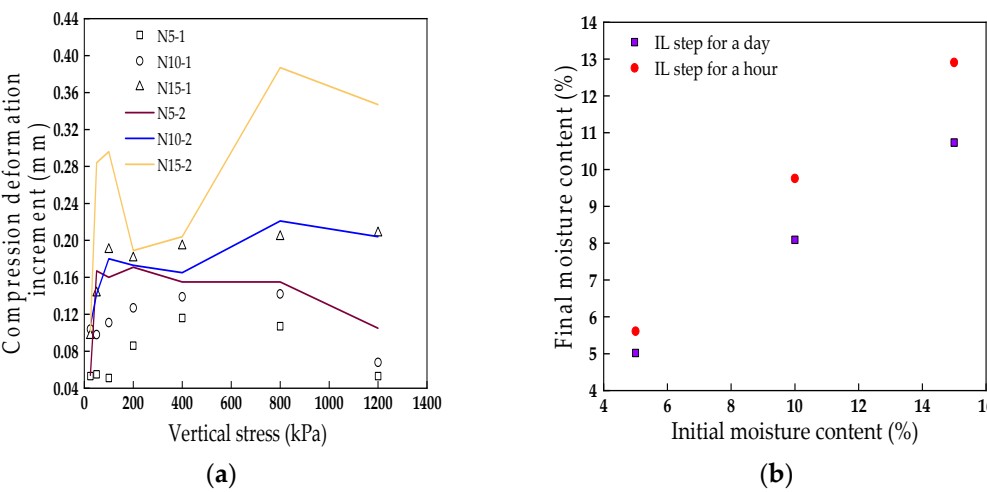

**Figure 25.** Comparison of IL tests with different loading duration: (**a**) comparison of compression deformation increment versus vertical stress; (**b**) comparison of moisture content after loading.

### 4.2.2. Resistivity

For the resistivity of the soil samples obtained via the IL tests, the resistivity usually decreases with the increase in stress. This is because when the instantaneous load is applied, the void ratio decreases, and the soil becomes denser; therefore, the gas in the pores is rapidly discharged, and the degree of free water connections in the pore space increases. As the deformation gradually stabilizes, the degree of pore water connection undergoes little change; thus, resistivity tends to stabilize as moisture content is unchanged [18–20]. In addition, the resistivity of soil samples decreases with increasing moisture content, which is because as the moisture content is high, the gas content in the soil pores is small, and the degree of free water connections in the pores increase; thus, resistivity decreases [19,22,35,43]. It seems that the aforementioned studies mainly explain the decrease in resistivity with the increase in stress and moisture content. However, the resistivity not only showed a decrease but also slightly increased for a loading step. It is believed that the increase in the resistivity of the samples is due to the loss of water during compression. Moreover, the variations in terms of moisture content shown in Figures 17 and 24 also validate the loss of water during compression.

In addition, the resistivity of the soil samples obtained when using the CRS tests depends on the effect of the strain rate. It can be seen in Figure 15 that as the void ratio of soil samples continuously decreases, the degree of pore water connection constantly increases and the amount of connected channel increases, leading to a decrease in resistivity. However, when the strain rate is $1 \times 10^{-6}$ s$^{-1}$, the resistivity of the soil sample slightly increases, even if the void ratio continues to decrease. Since soil sample drainage occurs when the void ratio decreases at the strain rate of $1 \times 10^{-6}$ s$^{-1}$, the amount of connected channel decreases with the loss of free water and resistivity increases. In addition, it can be seen in Figure 15 that the decrease in resistivity obtained using a strain rate of $1 \times 10^{-4}$ s$^{-1}$

is larger than that with a strain rate of $1 \times 10^{-5}\,\text{s}^{-1}$ for a given strain increment. This also indicates that the slower the loading rate, the more the drainage of the soil sample for a given compression deformation and the less the amount of connected channels.

## 5. Conclusions

This work studied the creep and resistivity properties of unsaturated compacted loess by using IL and CRS tests. The following conclusions were obtained.

(1)  $C_c$ and $C_\alpha$ of unsaturated compacted loess increase with the increase in vertical stress and moisture content. $C_\alpha$ increases linearly with $C_c$, which is distributed in a narrow range. $C_\alpha/C_c$ increases with the increase in moisture content.

(2)  For the IL tests, the void ratio decreases rapidly after the loading is imposed on the soil sample. The amount of electrical connected channels increases; thus, the resistivity of the soil sample decreases rapidly in the early loading stage. As the deformation of the soil sample stabilizes, the void ratio decreases slowly, and the number of connected channels decreases slightly. Importantly, the loss of water during compression dominates the increase in resistivity of the soil sample. For CRS tests, the resistivity of the soil sample decreases with increasing strain at a constant rate of strain. The resistivity rapidly decreases with the large strain rate. The resistivity decreases slowly at the latter loading stage than at the former stage for a given strain rate. However, as the strain rate is relatively small, the resistivity increases, intricating the significant impact of water loss on the resistivity. Meanwhile, the void ratio of soil samples slows down at the later loading stage due to the increased resistance to deformation of the soil particles.

(3)  In the NMR tests, it was found that there were three peaks in the $T_2$ distribution curves of unsaturated compacted loess, corresponding to different pore sizes. The spectral peaks in the $T_2$ curves shifted to the left when the vertical stress increased. The area enclosed by the spectral curve around the peak of the larger macropore sizes decreased while that of the smaller pore sizes increased, indicating that the larger macropores transformed into the small pores due to the compression of the soil samples. In addition, through the qualitative and quantitative analysis of the SEM images, it was found that with the increase in vertical stress, the pore area of soil samples decreased, the coarse particles were crushed, the fine particles were inlaid with each other and became agglomerates, the contact relationship between particles changed from point-to-point contact and edge to edge contact to face to face contact. The abundance and roundness of loess particles tended to build up with the increase in vertical stress and loading rate.

(4)  The primary and secondary compression indexes obtained when using the CRS tests have the same variation trends as those obtained using the IL tests. However, due to the fast loading rate, the drainage of the soil sample is less, and its corresponding moisture content is high. Hence, the compression parameters and compression deformation of the soil samples obtained using CRS tests are a little higher than those obtained when using the IL tests for a given vertical stress.

(5)  The moisture content has a significant impact on the creep characteristics of compacted loess. Therefore, it is necessary to strictly control the moisture content of the subgrade, roadbed, and slope in practical engineering to prevent the settlement of roadbeds and subgrades and landslide failure. Meanwhile, the duration of compression also has an impact on the deformation of compacted soil. Thus, it is necessary to consider the construction speed. For example, it is better to quickly process the ground through the means of compression or impact, but it is risky to quickly load the ground and slope during engineering construction, especially due to the fact that the moisture content of the loess is high. Moreover, the soil structure gradually stabilizes with the increase in vertical stress, and the post-construction creep deformation can be reduced through the pre-compression of the roadbed and subgrade. Therefore, this work could provide

a theoretical and experimental basis for improving ground stability and preventing landslide disasters.

**Author Contributions:** Conceptualization, Q.Y. and P.Q.; methodology, P.Q.; validation, Q.Y. and P.Q.; formal analysis, Q.Y. and P.Q.; resources, P.Q., Y.L. and Z.S.; data curation, Q.Y.; writing, original draft preparation, Q.Y. and P.Q.; writing, review and editing, Q.Y., P.Q., Y.L. and C.Y.; visualization, Q.Y., P.Q., Y.L., Z.S. and C.L.; supervision, P.Q., Y.L., Z.S. and C.L. All authors have read and agreed to the published version of the manuscript.

**Funding:** The research described in this paper was financially supported by the Natural Science Foundation of China (Grant No. 42177138 and 41907239) and the project was funded by the China Postdoctoral Science Foundation (2020M680909).

**Institutional Review Board Statement:** Not applicable.

**Informed Consent Statement:** Not applicable.

**Data Availability Statement:** The data presented in this study are available upon request from the corresponding author.

**Conflicts of Interest:** The authors declare no conflict of interest.

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
