# Peer review of "One-Dimensional Compressibility and Creep Characteristics of Unsaturated Compacted Loess Based on Incremental Loading and Constant Rate of Strain Methods"

_sustainability, doi:10.3390/su151813854_

Round 1
Reviewer 1 Report
This paper is quite long but is generally fine. The results are described according to standard geotechnical practices. There are a few things I would like the authors to do.
1. At the end of the abstract, there should be a statement about why this is important, what is the major application here?
2. Methods line 128 must be a lot clearer about the sample location - lat long needed. I am very confused about how many samples you have - fig 1 only have one sample., table 2 says 7 samples, 3.3.2 says 4 samples. How many? This needs to be much clearer. In the methods section you also need to name the loess Formation and talk about its age - age here has big implications for geochemical and geotechnical properties.
3. The conclusions just repeat what you have said already, what is the bigger picture here? comparison with other studies? representativeness of your samples? application of the results to engineering? Some wider context is needed here.
4. refs are ok but please format these correctly
None
Author Response
Response to Reviewer 1 Comments
Greatly thank you for your comments and advices. According to your comments and advices, we have conducted extensive revisions to our manuscript, the detailed answers are listed below.
Point 1: At the end of the abstract, there should be a statement about why this is important, what is the major application here?
Response 1: Thank you for your advices. We have added the content in the abstract "This investigation can provide theoretical base and experimental support for improving ground stability and preventing landslide disaster in the loess region."
Point 2: Methods line 128 must be a lot clearer about the sample location - lat long needed. I am very confused about how many samples you have - fig 1 only have one sample., table 2 says 7 samples, 3.3.2 says 4 samples. How many? This needs to be much clearer. In the methods section you also need to name the loess Formation and talk about its age - age here has big implications for geochemical and geotechnical properties.
Response 2: Thank you for your advice. The content is added as follows,
(1) In this study, the Q4 loess was extracted in Dongshan, Taiyuan. The soil in this region is widely used as subgrade and ground filling material. Importantly, the Dongshan region is a key protection region for landslides and geologic hazards in Shanxi Province. The depth of soil extraction is 1.2 meters below the ground surface. After collecting from the site, the soil material was sealed in bags to avoid mixing impurities during transportation.
(2) In order to distinguish the crushed loess material and compacted sample, the title of figure 1 was revised to “Figure 1. The particle distribution curve of the loess material.”
(3) We sincerely thank you for advices. We have renamed the sample number and added the following description: " There are totally 16 samples in this study, 11 samples are used for the IL tests (The loading duration of every stage for sample N5-1, N10-1, N15-1, N20, A-1 and A-2 is a day, while sample N5-1, N10-1, N15-1, B-1 and B-2 is a hour. Sample A-1, A-2, B-1 and B-2 are used for microstructure tests after compression), 5 samples are used for the CRS tests (Sample H-5, H10 and H15 are only used for compression test, while sample C-1 and C-2 are used for microstructure tests after CRS compression). The parameters of compacted samples are shown in Table 2.”
Point 3: The conclusions just repeat what you have said already, what is the bigger picture here? comparison with other studies? representativeness of your samples? application of the results to engineering? Some wider context is needed here.
Response 3: Thank you for your advices, we add the conclusion as follows,
"(5) The moisture content has a significant impact on the creep characteristics of compacted loess. Therefore, it is necessary to strictly control the moisture content of the subgrade, roadbed and slope in the practical engineering to prevent the settlement of the roadbed and subgrade and the landslide failure. Meanwhile, the duration of compression has also an impact on the deformation of compacted soil. Thus it is necessary to consider the construction speed construction. For example, it is better quickly process the ground by compression or impact, but it is risky for quickly loading to the ground and slope during engineering construction, especially as the water content of the loess is high. Moreover, the soil structure gradually stabilizes with the increase of vertical pressure, and the post-construction creep deformation can be reduced by pre-compression of the roadbed and subgrade. Therefore, this work could provide a theoretical and experimental base for improving the ground stability and preventing landslide disaster."
Point 4: refs are ok but please format these correctly.
Response 4: Thank you for your advice. We checked and revised the reference again in the light of the format requirement in the author guidelines.
Reviewer 2 Report
Generally, the manuscript is correctly written, clear and valuable for the study. The author studied one-dimensional creep properties of unsaturated compacted loess based on incremental loading (IL) and constant rate of strain (CRS) method in this work. However, the paper needs some improvement to enhance comprehension and meet the standard of the Sustainability journal.
1. In the Introduction, the author mentions a lot of previous work, so is there any deficiency in previous research? How does the author make new innovations on the basis of previous work?
2. Lines 40-41: “Due to the lack of rainfall and low water table in the northern regions, loess in these areas is typically unsaturated” Whether the sentence content is rigorous enough. How does this sentence relate to the context? Authors can refer to literature to complete the relevant background information (https://doi.org/10.3390/rs15030662; https://doi.org/10.3390/rs14102333).
3. The author does not account for the process and location of the soil sample collection, and whether the soil sample is representative of the region.
4. In the Figure 15b, the Resistivity versus strain of ω=5% was absent.
5. The font in Figure 16a is not fully displayed.
6. The pixels in Figure 17 are fuzzy and the resolution needs to be improved.
7. In “4.1. Compression properties addition”, is it necessary to consider the different properties of soil samples comparing the others' research results?
Author Response
Response to Reviewer 2 Comments
Greatly thank you for your comments and advices. According to your comments and advices, we have conducted extensive revisions to our manuscript, the detailed answers are listed below.
Point 1: In the Introduction, the author mentions a lot of previous work, so is there any deficiency in previous research? How does the author make new innovations on the basis of previous work?
Response 1: Thank you for your advices. We mention a lot of previous work so that we know what has been studied and inspire us to making new innovations. Based on the previous work, this study combines incremental loading with constant rate of strain to study the compressibility of loess, which could improve the reliability of the results by comparison and was better than only using a method. In addition, the change of soil water content was characterized by monitoring the change of electrical resistivity during the compression process, and the macroscopical compression creep deformation was further verified by microscopical changes obtained from nuclear magnetic resonance and scanning electron microscopy tests, which made the research more persuasive. Notably, we validated that quick compression can increase deformation from macroscopical and microscopical points. All of these studies mark new innovations.
Point 2: Lines 40-41: “Due to the lack of rainfall and low water table in the northern regions, loess in these areas is typically unsaturated” Whether the sentence content is rigorous enough. How does this sentence relate to the context? Authors can refer to literature to complete the relevant background information (https://doi.org/10.3390/rs15030662; https://doi.org/10.3390/rs14102333).
Response 2: Thank you for your advices. We have added more references to support the background information. “In addition, landslide failure accelerates as the moisture content of the soil increases during rainfall, posing a significant threat to roads, rivers, and human life.[4, 5]”
([4] Wang, L.; Qiu, H.; Zhou, W.; Zhu, Y.; Liu, Z.; Ma, S.; Yang, D.; Tang, B. The post-failure spatiotemporal deformation of certain translational landslides may follow the pre-failure pattern. Remote Sensing 2022, 14, (10), 2333.
[5] Ma, S.; Qiu, H.; Zhu, Y.; Yang, D.; Tang, B.; Wang, D.; Wang, L.; Cao, M. Topographic Changes, Surface Deformation and Movement Process before, during and after a Rotational Landslide. Remote Sensing. 2023, 15, 662.)
Point 3: The author does not account for the process and location of the soil sample collection, and whether the soil sample is representative of the region.
Response 3: Thank you for your advice. We have added to section 2.1.1 Basic properties of loess. The details are shown as follows:" In this study, the Q4 loess was extracted in Dongshan, Taiyuan. The soil in this region is widely used as subgrade and ground filling material. Importantly, the Dongshan region is a key protection region for landslides and geologic hazards in Shanxi Province. The depth of soil extraction is 1.2 meters below the ground surface. After collecting from the site, the soil material was sealed in bags to avoid mixing impurities during transportation."
Point 4: In the Figure 15b, the Resistivity versus strain of ω=5% was absent.
Response 4: Thank you for your advice. We have revised Fig.15. Fig.15a is the resistivity versus strain of ω=5%. Fig.15b is the resistivity versus strain of ω=10% and ω=15%. Because the electrical resistivity value has a larger difference so that it is not clear to distinguish the variation trends of the resistivity versus strain curves of ω=10% and ω=15%. Thus, we separate them.
Point 5: The font in Figure 16a is not fully displayed.
Response 5: Thank you for your advice. The figure 16a is revised as follows,
Point 6: The pixels in Figure 17 are fuzzy and the resolution needs to be improved.
Response 6: Thank you for your advice. We have improved the image resolution.
- Point 7: In “4.1. Compression properties addition”, is it necessary to consider the different properties of soil samples comparing the others' research results?
Response 7: Thank you for your advice. By comparison of the others' research results, we want to know the variation of the compression parameters and whether it is consistent with other researches. If they are consistent, it is validated that the soil has the same variation with the others’ soil. If they are different, we want to further explore what causes the difference. Therefore, we think that it is necessary to compare with the others' research results.
Reviewer 3 Report
Abstract
An overview of the papers and a summary of the One-dimensional creep characteristics of unsaturated compacted loess based on incremental loading and constant rate of strain approaches are included in the abstract section. However, this study has examined the “one-dimensional creep properties of unsaturated compacted loess using the incremental loading and constant rate of strain methods”. Additionally, the benefits and drawbacks of adopting “one-dimensional creep qualities with incremental loading and constant rate of strain” have also been discussed. Hence, the study includes illustrations of the application approach and a discussion of the main results.
Introduction
The introductory section has shed light on the impact of “Loess, a loose aeolian deposit of yellowish silt-sized dust”. Additionally, progress in studying soil creep properties also has been illustrated in the study. It has been identified that researchers have used the constant rate of strain test method to examine soil creep characteristics. However, this study examines the microstructure, resistivity, and one-dimensional creep properties of unsaturated compacted loess. The primary compression index, secondary compression index, and their ratio were each computed by using two separate loading techniques
Literature review
The literature review on the study has been determined in 3rd section which defines the impact of “one-dimensional creep qualities with incremental loading and constant rate of strain”. The study has demonstrated the microcomputer-controlled electronic testing device has the advantage of a simple design and operation in the CRS loading frame. Additionally, it includes drawbacks like a lengthy test period, discrete data collection locations, and test-related human intervention. Moreover, the review paper defines the procedure and benefits of using “one-dimensional creep qualities with incremental loading and constant rate of strain”.
Materials and Method
The methods of making the research hypothesis and procedure of creating questionnaires for surveying the topic impact of “one-dimensional creep qualities with incremental loading and constant rate of strain” have been identified. The study has examined “Test soil sample and preparation”, “Soil sample preparation”, and “Experimental protocol”. Additionally, statistical research and analysis also have been done to sustain the quality of the research paper.
Result and Discussion
The research on the study has provided information on the usage of the method. The result of Compression deformation, Resistivity properties, CRS test, and Microstructure test have been analysed in this section. On a contradictory note, Compression properties, Comparison of incremental loading and CRS test and Resistivity have been discussed in the study. Additionally, the research paper has evaluated the correlation between the one-dimensional creep properties of unsaturated compacted loess using “incremental loading and constant rate of strain methods”.
Conclusion
The study has concluded the impacts of using one-dimensional creep properties of unsaturated compacted loess using “incremental loading and constant rate of strain methods”. Additionally, it also concluded that the different methods of examining and using the methods have various impacts. Hence, the study concluded the positive effect of using the technology.
Comments for authors
Instead of concentrating on statistical perspective research, it is advised that the authors enrich the theoretical and emotional perspectives.
Minor editing of English language required
Author Response
Response to Reviewer 3 Comments
Greatly thank you for your comments and advices. According to your comments and advices, we have conducted extensive revisions to our manuscript, the detailed answers are listed below.
Point 1: Abstract
An overview of the papers and a summary of the One-dimensional creep characteristics of unsaturated compacted loess based on incremental loading and constant rate of strain approaches are included in the abstract section. However, this study has examined the “one-dimensional creep properties of unsaturated compacted loess using the incremental loading and constant rate of strain methods”. Additionally, the benefits and drawbacks of adopting “one-dimensional creep qualities with incremental loading and constant rate of strain” have also been discussed. Hence, the study includes illustrations of the application approach and a discussion of the main results.
Response 1: Thank you for your comments.
Point 2: Introduction
The introductory section has shed light on the impact of “Loess, a loose aeolian deposit of yellowish silt-sized dust”. Additionally, progress in studying soil creep properties also has been illustrated in the study. It has been identified that researchers have used the constant rate of strain test method to examine soil creep characteristics. However, this study examines the microstructure, resistivity, and one-dimensional creep properties of unsaturated compacted loess. The primary compression index, secondary compression index, and their ratio were each computed by using two separate loading techniques.
Response 2: Thank you for your comments. Electrical resistivity method is a nondestructive method that was used to characterize the change of moisture content to obtain more information about the sample. One-dimensional creep properties of unsaturated compacted loess are macroscopic deformation, which were significantly influenced by moisture content. Meanwhile microstructure could deeply explain how the creep happened. All the test methods can be mutual confirmation so that increase the reliability of results.
Point 3: Literature review
The literature review on the study has been determined in 3rd section which defines the impact of “one-dimensional creep qualities with incremental loading and constant rate of strain”. The study has demonstrated the microcomputer-controlled electronic testing device has the advantage of a simple design and operation in the CRS loading frame. Additionally, it includes drawbacks like a lengthy test period, discrete data collection locations, and test-related human intervention. Moreover, the review paper defines the procedure and benefits of using “one-dimensional creep qualities with incremental loading and constant rate of strain”.
Response 3: Thank you for your comments. Incremental loading method has drawbacks like a lengthy test period, discrete data collection locations, and test-related human intervention. As the microcomputer-controlled electronic testing device has the advantage of a simple design and operation in the CRS loading frame.
Point 4: Materials and Method
The methods of making the research hypothesis and procedure of creating questionnaires for surveying the topic impact of “one-dimensional creep qualities with incremental loading and constant rate of strain” have been identified. The study has examined “Test soil sample and preparation”, “Soil sample preparation”, and “Experimental protocol”. Additionally, statistical research and analysis also have been done to sustain the quality of the research paper.
Response 4: We feel great thanks for your professional review work on our article.
Point 5: Result and Discussion
The research on the study has provided information on the usage of the method. The result of Compression deformation, Resistivity properties, CRS test, and Microstructure test have been analyzed in this section. On a contradictory note, Compression properties, Comparison of incremental loading and CRS test and Resistivity have been discussed in the study. Additionally, the research paper has evaluated the correlation between the one-dimensional creep properties of unsaturated compacted loess using “incremental loading and constant rate of strain methods”.
Response 5: We feel great thanks for your professional review work on our article.
Point 6: Conclusion
The study has concluded the impacts of using one-dimensional creep properties of unsaturated compacted loess using “incremental loading and constant rate of strain methods”. Additionally, it also concluded that the different methods of examining and using the methods have various impacts. Hence, the study concluded the positive effect of using the technology.
Response 6: Thanks for your comments. We sincerely appreciate the valuable comments.
Point 7: Comments for authors
Instead of concentrating on statistical perspective research, it is advised that the authors enrich the theoretical and emotional perspectives.
Response 7: Thank you for your advices. The conclusion further adds the application about this study and we actually conduct much deeper studies, such as time-dependent constitutive model and multiple influence factors.
Point 8: Comments on the Quality of English Language
Minor editing of English language required.
Response 8: Thank you for your advice. The language is edited again.
Round 2
Reviewer 2 Report
This manuscript has been improved according my comments. I believe it should be accepted in current form.